# Distribution and Prevalence of *Theileria orientalis* Genotypes in Adult Lactating Dairy Cows in South West Region of Western Australia

**DOI:** 10.3390/pathogens12010125

**Published:** 2023-01-12

**Authors:** Chi-Cheng Leong, Charlotte L. Oskam, Amanda D. Barbosa, Joshua W. Aleri

**Affiliations:** 1School of Veterinary Medicine, College of Science, Health, Engineering and Education, Murdoch University, 90 South Street, Murdoch, WA 6150, Australia; 2Centre for Biosecurity and One Health, Harry Butler Institute, Murdoch University, 90 South Street, Murdoch, WA 6150, Australia; 3CAPES Foundation, Ministry of Education of Brazil, Brasilia-DF 70040-020, Brazil; 4Centre for Animal Production and Health, Future Foods Institute, Murdoch University, 90 South Street, Murdoch, WA 6150, Australia

**Keywords:** bovine anaemia, genotype, Haemaphysalis longicornis, Theileria orientalis

## Abstract

Bovine anaemia caused by *Theileria orientalis* group (BATOG) causes significant production and economic losses in Australia’s cattle industry. The pathogenic *T. orientalis* genotypes reported in Australian cattle are type 1 (Chitose) and type 2 (Ikeda). The present study aimed to determine the prevalence and distribution of *T. orientalis* genotypes in adult lactating cows in Western Australia (WA) dairy herds. A total of 100 whole blood samples from lactating cows from 10 farms were obtained and screened for *T. orientalis* using polymerase chain reaction (PCR). Sanger sequencing was subsequently used to characterise *T. orientalis* genotypes isolated from positive samples. A total of thirteen cows (13%; 95% CI: 7.1–21.2%) were positive for *T. orientalis*, and six out of ten farms (60%; 95% CI: 26.2–87.8%) housed at least one *T. orientalis*-positive cow. The distribution of *T. orientalis* was found to be wide and dense in the South west region of WA and the southern coast of WA. The predominant *T. orientalis* genotype identified was Ikeda (n = 11, 11%; 95% CI: 5.6–18.8%), while the Buffeli genotype was identified in WA for the first time, albeit at a low prevalence (n = 1, 1%; 95% CI: 0.0–5.4%). This study has provided useful epidemiological evidence on the prevalence and distribution of *T. orientalis* in adult lactating dairy cows in WA dairy farms, and on the importance of conducting widespread surveillance programs for the understanding of BATOG in WA.

## 1. Introduction

*Theileria orientalis* is a blood-borne intracellular protozoan parasite, transmitted to cattle by Australian bush ticks (syn. Asian long-horned ticks, New Zealand cattle ticks), *Haemaphysalis longicornis*, causing bovine anaemia caused by *Theileria orientalis* group (BATOG) [1,2]. Clinical signs of BATOG include severe anaemia, pyrexia, lethargy, lack of appetite, jaundice, diarrhoea, exercise intolerance, abortions, and mortality [3,4]. Currently, *T. orientalis* is classified into 11 genotypes (Types 1–8 and N1–N3) based on phylogenetic analysis of the major piroplasm surface protein (MPSP) gene sequence [1,5]. To date, five genotypes have been reported in Australia: *T. orientalis* type 1 (Chitose), type 2 (Ikeda), type 3 (Buffeli), type 4 (type C) and type 5 [1,6,7,8]. Of these, type 1 (Chitose) and type 2 (Ikeda) have been associated with disease outbreaks in Australian cattle [9,10]. 

Since 2006, more than 500 clinical outbreaks of BATOG in Australian cattle have been identified in New South Wales (NSW), Queensland (Qld), Victoria (Vic), Sothern Australia (SA) and Western Australia (WA). Due to the increase in the number, severity, and distribution of clinical cases of BATOG over the past decade, BATOG has become one of the most economically significant diseases of cattle in Australia, especially in dairy cattle [8]. It has been estimated that the annual economic loss caused by a single lactating cow suffering from clinically severe BATOG is AUD 202 due to a significant milk-production loss, whilst the loss caused by the death of cattle is approximately AUD 1800 per head [11]. 

In WA, BATOG was first detected in the Shire of Denmark in the Great Southern region in 2013, with additional clinical cases being found in the neighbouring Shires of Capel, Harvey, and Albany in 2014 and 2015 [4,12]. The prevalence of *T. orientalis* recorded in Denmark in 2015 was 34%, with Ikeda being identified as the dominant genotype (23%) [12]. Despite these recent reports, there remains limited data on the epidemiology of BATOG in WA. To address this, this study aimed to investigate the prevalence, genetic diversity, and geographical distribution of *T. orientalis* in dairy farms in the South West region of WA.

## 2. Materials and Methods

### 2.1. Ethics Approval 

Data collection for this study was approved by the Animal Ethics Committees of Murdoch University (approval no. 2019/047).

### 2.2. Study Area

This study was conducted in the South West region of WA. Dairy farms are predominantly located to the south-west of Perth (capital city of WA). This region has a Mediterranean climate with an average annual precipitation of approximately 900 mm, being well-suited conditions for *Haemaphysalis longicornis* ticks [13].

### 2.3. Study Design and General Data Collection

This study was part of a broader project investigating calf health and welfare in WA with samples being collected from April 2019 to June 2020 [14]. This was a cross-sectional study where study farms were visited once to collect blood samples from healthy adult lactating dairy cows. All blood samples were obtained from the tail vein of each animal and then immediately placed into an EDTA tube, labelled, and stored at Murdoch University. 

### 2.4. Study Populations and Study Farms

A convenience sample of 34 registered dairy producers was invited to participate via email. Participation was voluntary, and no incentives were provided. A total of 29 farms were sampled, of which 10 were selected based on their distribution for the purpose of this study.

### 2.5. Sample Size Estimations and Selection of Study Subjects

The initial sample size of 29 animals per farm was chosen based on our previous study [14]. Assuming a 50% prevalence of pathogens with a 95% confidence interval while allowing for a 5% error rate, a total of 784 animals (20–25 animals per farm) was deemed appropriate. For each farm, random selection was conducted by first identifying all the adult lactating cows that met the selection criteria and numbering them serially as 1, 2, 3 to s, where s was the last serial number depending on the total number of adults. Using this database, a total of 25 random numbers representing the animals to be sampled were generated in Excel spread sheets. In the present study, a 50% seroprevalence with a 95% confidence interval while allowing for a 10% error rate was assumed, hence a total of 97 animals was calculated using EpiTools sample size calculations [15]. Therefore, of the 20–25 animals sampled per farm, 10 were further randomly selected from each selected farm by randomly generating 10 numbers in Excel.

### 2.6. DNA Extraction and Polymerase Chain Reaction Assays (PCR)

Genomic DNA (gDNA) was extracted from 200 µL blood samples using the MasterPure™ DNA Purification Kit for Blood Version II (EPICENTRE^®^ Biotechnologies, Madison, WI, USA), according to the manufacturer’s instructions. DNA samples were screened by qPCR targeting the 16S mitochondrial DNA using the primer sets, 16Smam1 and 16Smam2, to assess the quality of extractions [16]. Each reaction was performed in a final volume of 25 μL containing the following: 12.5 μL of Power SYBR^®^ Green Master Mix, 1 μL of forward primer, 1 μL of reverse primers, 8.5 μL of DNA-free water, and 1 μL of gDNA, subjected to the following PCR conditions: a preliminary cycle at 95 °C for 5 min followed by 40 cycles at 95 °C for 30 s, 21 and 57 °C for 30 s, and 72 °C for 30 s. DNA samples were also screened by qPCR targeting the 18S rRNA gene of Piroplasmida organisms using the primer sets 18SApiF and 18SApiR, as previously described [17]. The same samples were then screened by qPCR targeting the *T. orientalis* MPSP gene using the primer sets MPSP_F and MPSP_R [10]. Each reaction was performed in a final volume of 25 µL containing the following: 12.5 µL of Power SYBR^®^ Green Master Mix, 1 µL of forward primer, 1 µL of reverse primers, 8.5 µL of DNA-free water, and 2 µL of gDNA. Assays were carried out subjected to the following PCR thermos-cycling conditions: an initial hold at 95 °C for 5 min, followed by 45 cycles at 95 °C for 30 s, and at 58 °C for 30 s. Finally, all extracted DNA was screened by cPCR targeting the *T. orientalis* MPSP gene using the primer sets MPSP_F_Mod and MPSP_R. Each reaction was performed in a final volume of 25.35 µL containing the following: 17.5 µL of DNA-free water, 2.5 µL of 1× (1.5 mM) KAPA Taq Buffer + dye (with MgCl2), 1 µL of 1.0 mM KAPA MgCl2, 1 µL of 10 µM MPSP_F_Mod forward primer, 1 µL of 10 µM MPSP_R reverse primer, 0.25 µL of 0.25 mM dNTPs (Fisher Biotech), 0.1 µL / 0.5 U KAPA Taq and 2 µL of gDNA. Assays were carried out subjected to the following PCR thermos-cycling conditions: an initial hold at 95 °C for 2 min, followed by 45 cycles at 95 °C for 30 s, 55 °C for 30 s, and 72 °C for 1 min, with a final extension at 72 °C for 5 min for genotype identification. Primers used in the study are listed in Table 1.

### 2.7. DNA Purification and Sanger Sequencing

Products amplified from qPCR assays were purified using QIAquick PCR Purification Kit (50) (Qiagen, Hilden, Germany) according to the manufacturer’s instructions. Products amplified from cPCR assays were run on a 1% agarose gel (Fisher Biotech, Wembley, WA, Australia) containing 4 µL of SYBR^®^ Safe (Invitrogen^TM^, Waltman, MA, USA). Gel bands were visualised with an Ultra-Violet 25 transilluminator (BioRad, Hercules, CA, USA) and those corresponding to the expected length were excised and purified using an in-house filter-tip method as previously described [19]. All purified PCR products were sequenced in both forward and reverse directions using an ABI Prism^TM^ Terminator Cycle Sequencing kit (Applied Biosystems, Foster City, CA, USA) on an Applied Biosystem 3730 DNA Analyzer.

### 2.8. Genotype Identification and Phylogenetic Analysis

Sequences obtained in this study were imported into Geneious v10 [20] and firstly compared against the nucleotide database of the National Center for Biotechnology Information (NCBI), using the BLAST tool. The species identity was confirmed against the highest percentage of BLAST results with a significant query cover. Alignments with reference sequences acquired from GenBank were built using MAFFT [21]. Aligned sequences were then imported into MEGA11 to determine the best nucleotide substitution model based on the Bayesian Information Criterion (BIC) [22]. Phylogenetic trees were then generated using the maximum-likelihood method with two distantly related *Theileria* species (*Theileria parva* and *Theileria annulata*) as an outgroup, based on 1000 bootstrap replications.

## 3. Results

### 3.1. Prevalence and Distribution of Theileria orientalis 

Of the ten farms sampled, *T. orientalis* was detected on six (60%, 95% CI: 26.2–87.8%). Of the 100 animals sampled, 13 (13%, 95% CI: 7.1–21.2%) were positive for *T. orientalis*. The distribution of *T. orientalis* detected in the present study was widespread throughout the South West region, and dense in the western south-west corner of WA and the southern coast of WA. The locations of the farms and the distribution of *T. orientalis* are depicted in Figure 1. The number of *T. orientalis*-positive animals found within farms ranged from one to four. The within-farm prevalence of *T. orientalis* is presented in Table 2. Farms located on the southern coast of WA had a slightly higher prevalence (30–40%) of *T. orientalis* than the farms located in the western South West region of WA (10–20%). 

### 3.2. Prevalence of Theileria orientalis Genotypes 

An Ikeda prevalence of 11% (95% CI: 5.6–18.8%) and a Buffeli prevalence of 1% (95% CI: 0.0–5.4%) were recorded. Only one genotype was found in one single farm positive for *T. orientalis*. The within-farm *T. orientalis* genotypes detected are detailed in Table 2. In addition, T. *orientalis* Ikeda genotype and Buffeli genotype could be discriminated by melt-curve analysis of the MPSP qPCR assays as shown in Figure 2.

### 3.3. Phylogenetic Analysis

A phylogeny of *T. orientalis* genotypes based on MPSP nucleotide sequences (692 bp) generated using the maximum-likelihood method (ML) of Tamura 3-parameter + discrete gamma distribution (K92+G) algorithms is shown in Figure 3. The topology of the phylogenetic tree shows the MPSP nucleotide sequences from this study clustered in two separate clades. The identical MPSP sequences formed a monophyletic clade with Ikeda genotypes, separated from the *T. orientalis* genotype within other clades, and was supported by a 98% bootstrap confidence value. One sample clustered within the monophyletic clade of Buffeli was separated from the *T. orientalis* genotype of other clades and was supported by a 100% bootstrap confidence value.

## 4. Discussion

The present study provides the first insight into the distribution and prevalence of *T. orientalis* genotypes in adult lactating dairy cows across dairy farms in WA. Overall, *T. orientalis* was detected in 60% of the farms and its prevalence was 13%, with an Ikeda genotype prevalence of 11% and a Buffeli genotype prevalence of 1%.

The distribution of *T. orientalis*-positive farms in the present study was widespread throughout the South West region of WA, especially in the western South West region and the southern coast. According to data from the Department of Primary Industries and Regional Development (DPIRD) Diagnostics and Laboratory Services (DDLS) (formerly DAFWA Animal Health Laboratories), clinical cases were diagnosed between 2012 and 2015 in the Shires of Capel, Harvey, Albany, and Denmark [12]. Adding to previous findings, *T. orientalis* has been detected also in Busselton and Dardanup in the present study, which constitutes additional supporting evidence for the widespread distribution of this parasite across the greater southwest of WA. This rapid spread could be caused by regular intra-state cattle movement, and records of intra-state cattle movement into and out of the Shire of Denmark were found [4]. In WA, BATOG was first reported in the Shire of Denmark [4]. However, there are no records of cattle importation from other states of Australia into Denmark in the last eight years according to the National Livestock Information System (NLIS) database [4]. It is likely that *T. orientalis*-infected cattle were imported from other states to the locations where *T. orientalis* was detected in this study, prior to being transported to Denmark. While there are no requirements for moving cattle around southwest WA, tick dipping is required if cattle are moving from northern WA into southern WA due to the Australian cattle tick, *Rhipicephalus australis*. *Rhipicephalus australis* is associated with the transmission of *Anaplasma marginale*, *Babesia bigeminia* and *Babesia bovis*, causing tick fever [23]. Furthermore, the *T. orientalis* prevalence found in this study (13%, 95% CI:7.1–21.2%) was slightly lower than has been previously observed in The Shire of Denmark (34%) [13]. The nature of the farms that were surveyed in the current study (i.e., farms that had not been reported as being associated with clinical outbreaks), made comparisons challenging because other studies had specifically sampled cattle where clinical outbreaks had occurred. However, in a survey by Eamens et al. [9], temporal prevalence of *T. orientalis* in NSW (23.7%), Qld (56.8%), and Vic (34.0%) was determined in ‘healthy’ (or asymptomatic) cattle using PCR through laboratory submissions. The prevalence of *T. orientalis* presented in this study is lower compared to other states [9]. Differences in prevalence could also be due to the different sensitivity of molecular tests used and variable levels of parasitaemia. 

In addition, *T. orientalis* was previously distributed mostly in coastal regions with high rainfall. The first Theileria infection in SA was recorded in a high rainfall area in 2012 with minor BATOG outbreaks occurring in 2014, 2016 and 2017 [7]. An increasing *H. longicornis* population size has been reported due to elevated rainfall, especially in the southeast region of SA, resulting in widespread BATOG in SA [24]. *H. longicornis* was identified in the BATOG outbreaks in Albany, indicating that *H. longicornis* are a common vector in the Shires of Albany and Demark, which are geographically close to each other. *H. longicornis* ticks were also reported in the Shire of Harvey [12]. However, knowledge of the geographical distribution of *H. longicornis* ticks in WA is lacking. 

The Ikeda genotype detected in the present study was significantly more common than the Buffeli genotype, which is in agreement with previous studies [4,12]. Although the Ikeda genotype is considered highly pathogenic, the cattle positive for Ikeda genotype in this study did not exhibit clinical signs of disease. There are several possible scenarios which could explain the lack of clinical signs [6,25,26]. Firstly, the high prevalence of the Ikeda genotype detected in this study could be indicative of the presence of a genetically less pathogenic variant of the Ikeda genotype with low parasitaemia [6]. Secondly, parasitaemia can persist within herds for at least five months [25]. Thirdly, a high level of population immunity is present, representing the achievement of enzootic stability [6]. Fourthly, the cattle have recovered and remain asymptomatic carriers [26].

A novel finding of a single Buffeli genotype was detected in this study. To our knowledge, Buffeli genotypes were not detected in any previous study conducted in WA using Buffeli genotype-specific PCR assays [4,12]. The Buffeli genotype is generally considered to be avirulent, slow to develop, and with low parasitaemia [27]. In addition, animals with Buffeli infections generally exhibit either no or mild clinical signs [28], and it is unlikely that asymptomatic animals would be tested for *T. orientalis*. Therefore, the Buffeli genotype may have been previously undetected due to the low parasitaemia and the lack of clinical signs of infection.

Mixed infections of two or more genotypes of *T. orientalis* have commonly been detected in NSW, Qld, Vic and SA [1,7,29]. Considering the detection of two genotypes of *T. orientalis* in the present study, there might be a possibility for potential mixed infections in WA. However, mixed genotype populations were not able to be detected using cPCR and Sanger sequencing in the present study.

Despite the small sample size this study has provided useful epidemiological evidence on the prevalence and distribution of *T. orientalis* in the southwest dairying region of Western Australia. There is need for continued active and passive surveillance to further understand the relationship of the pathogens occurring in clinical and subclinical cases.

## 5. Conclusions

This study revealed the distribution of *T. orientalis* in dairy farms in the greater South West region of WA is widespread, especially in the western southwest corner and on the southern coast. The *T. orientalis* prevalence was 13%, while the Ikeda genotype was the prominent genotype. Future research on the characterisation and distribution of ticks on the west coast of WA will provide useful epidemiological information.

## Figures and Tables

**Figure 1 pathogens-12-00125-f001:**
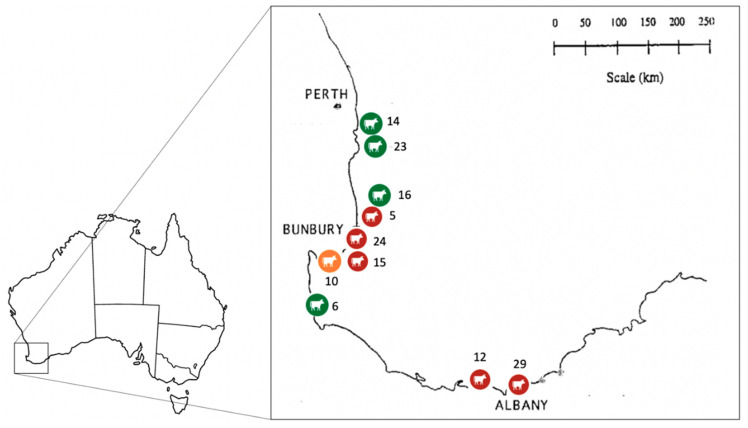
Map of the infected farms (red: Ikeda; orange: Buffeli) and non-infected farms (green). Numbers are farm identifiers. 5: Brunswick; 6: Forest Grove; 10: Busselton; 12: Denmark; 14: Oldbury; 15: Boyanup; 16: Harvey; 23: North Dandalup; 24: Dardanup; 29: Albany.

**Figure 2 pathogens-12-00125-f002:**
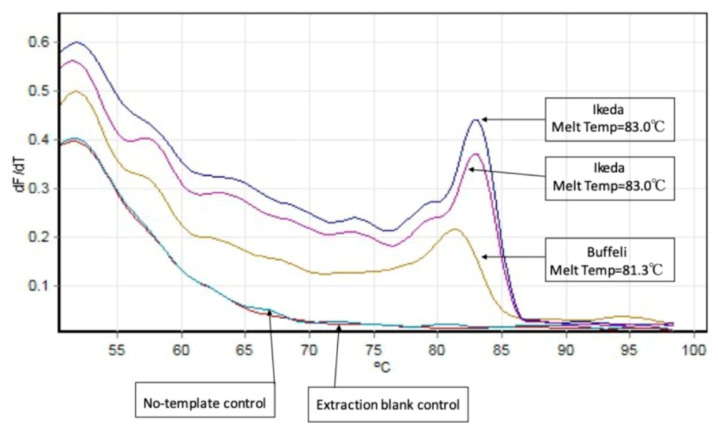
A melt curve analysis of MPSP qPCR assays comparing the melting temperature of Ikeda and Buffeli samples.

**Figure 3 pathogens-12-00125-f003:**
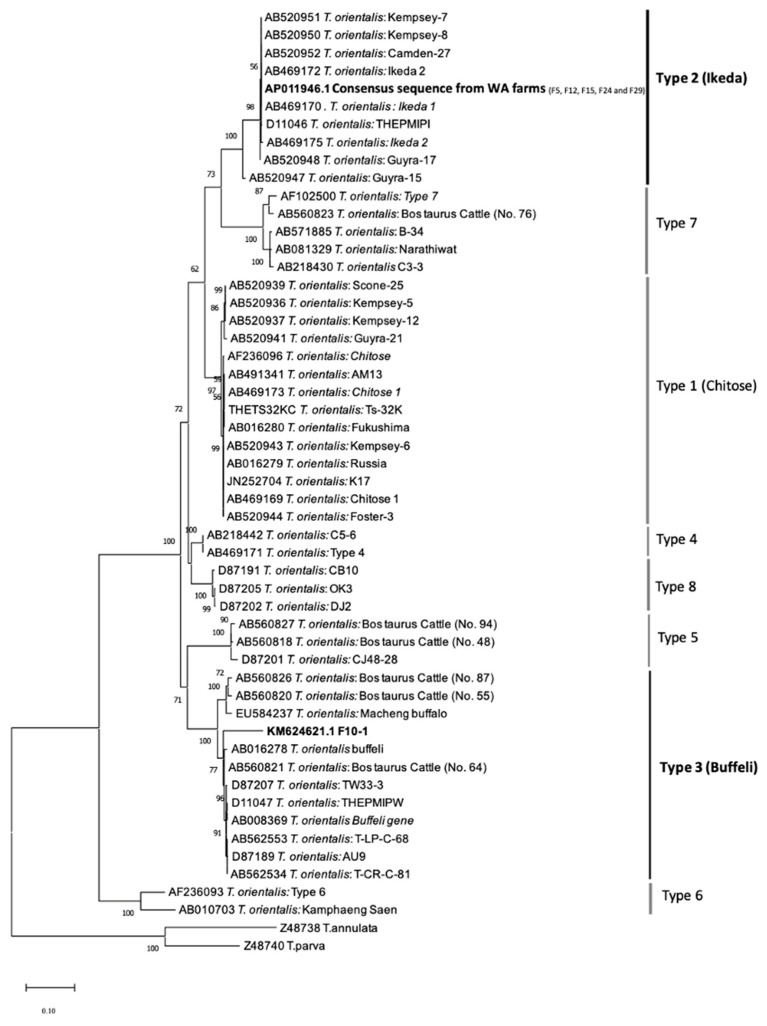
Phylogeny of bovine *Theileria orientalis* genotypes based on MPSP nucleotide sequences (692 bp) generated using the maximum-likelihood method (ML) and Tamura 3-parameter + discrete gamma distribution (K92+G) algorithms. *T. annulata* and *T*. *parva* were used as the outgroup. Bootstrap support level (>50%) from 1000 replicates is indicated at the left of the supported node. The scale bar indicates genetic distance. The sequences generated in this study are highlighted in bold.

**Table 1 pathogens-12-00125-t001:** Primers used for PCR analysis in the study.

Gene	Target	Primer	Sequence (5′–3′)	Size (bp)	Reference
16S	Mitochondrial DNA	16Smam1 16Smam2	GGTTGGGGTGACCTCGGA CTGTTATCCCTAGGGTAACT	~140	[16]
18S	Piroplasms	18SApiF 18SApiR	ACGAACGAGACCTTAACCTGCTA GGATCACTCGATCGGTAGGAG	~300	[17]
MPSP (short)	*Theileria orientalis* spp.	MPSP_F MPSP_R	TGCTATGTTGTCCAAGAGAACG TGTGAGACTCAATGCGCCTAGA	~300	[10]
MPSP (long)	*Theileria orientalis* spp.	MPSP_F_Mod MPSP_R	GCAAACAAGGATTTGCACGC TGTGAGACTCAATGCGCCTAGA	~847	Modified from [10,18]

**Table 2 pathogens-12-00125-t002:** Summary of the prevalence of *Theileria orientalis* within farm and *Theileria orientalis* genotype identification.

Farm	MPSP qPCR (Short)		MPSP cPCR (Long)		
	Number of Animals Positive for *T. orientalis* (Prevalence %)	95% CI	Number of Animals Positive for *T. orientalis* (Prevalence %)	95% CI	Genotype(s)
5	1 (10%)	0.3–44.5%	1 (10%)	0.3–44.5%	Ikeda
10	1 (10%)	0.3–44.5%	1 (10%)	0.3–44.5%	Buffeli
12	4 (40%)	12.2–73.8%	3 (30%)	6.7–65.2%	Ikeda
15	2 (20%)	2.5–55.6%	2 (20%)	2.5–55.6%	Ikeda
24	1 (10%)	0.3–44.5%	1 (10%)	0.3–44.5%	Ikeda
29	4 (40%)	12.2–73.8%	4 (40%)	12.2–73.8%	Ikeda
Total	13 (13%)	7.1–21.2%	12 (12%)	6.4–20.0%	Buffeli + Ikeda

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
