# Peer review of "Distribution and Prevalence of Theileria orientalis Genotypes in Adult Lactating Dairy Cows in South West Region of Western Australia"

_pathogens, 2023, doi:10.3390/pathogens12010125_

Round 1
Reviewer 1 Report
This manuscript on "Distribution and prevalence of Theileria orientalis genotypes in adult lactating dairy cows in Western Australia" is solid and well written. The authors have completed a considerable amount of work.
1. Please cross check the references in the list with the references cited in the text. Following references were not found in the text:
Taylor, P. G. (1996). Reproducibility of Ancient DNA Sequences from Extinct Pleistocene Fauna. Molecular Biology and Evolution, 13(1), 308 283-285.
Kawazu, S.-i., Sugimoto, C., Kamio, T., & Fujisaki, K. (1992). Analysis of the genes encoding immunodominant piroplasm surface 285 proteins of Theileria sergenti and Theileria buffeli by nucleotide sequencing and polymerase chain reaction. Molecular and bio-286 chemical parasitology, 56(1), 169-175.
Cufos, N., Jabbar, A., de Carvalho, L. M., & Gasser, R. B. (2012). Mutation scanning-based analysis of Theileria orientalis populations 261 in cattle following an outbreak. Electrophoresis, 33(13), 2036-2040.
2. Lines 41-44: Certain relevant references are missing in the introduction and discussion. Particularly those papers that reported first outbreaks in other states in Australia and the first discovery of certain T. orientalis genotypes.
E.g., Gebrekidan, H., Gasser, R.B., Perera, P.K., McGrath, S., McGrath, S., Jabbar, A., 2015. Investigating the first outbreak of oriental theileriosis in cattle in South Australia using multiplexed tandem PCR (MT-PCR). Ticks and Tick-borne Diseases 6, 574-578.
3. Since the sampling was restricted to the south-west region of WA, authors can consider including that to the title rather than 'Western Australia'.
4. Line 85: Why didn't the authors use a study which investigated the prevalence of Theileria orientalis in WA or any other state to calculate the sample size for the current study?
5. Figure 3. Phylogeny of bovine Theileria orientalis genotypes...
6. Consider stating the location of each sequence in Figure 3 rather than the isolate name.
7. Line 187: Is it across WA or south-west region of WA?
8. Line 209: Babesia bigemina. Please check spellings.
9. Lines 218-220: Add references.
10. Can authors discuss why mixed infections were not detected? Mixed infections of 2 or more genotypes of T. orientalis have been commonly reported in other states of Australia.
Author Response
Reviewers 1
Comments and Suggestions for Authors
This manuscript on "Distribution and prevalence of Theileria orientalis genotypes in adult lactating dairy cows in Western Australia" is solid and well written. The authors have completed a considerable amount of work.
- Please cross check the references in the list with the references cited in the text. Following references were not found in the text:
Taylor, P. G. (1996). Reproducibility of Ancient DNA Sequences from Extinct Pleistocene Fauna. Molecular Biology and Evolution, 13(1), 308 283-285.
Kawazu, S.-i., Sugimoto, C., Kamio, T., & Fujisaki, K. (1992). Analysis of the genes encoding immunodominant piroplasm surface 285 proteins of Theileria sergenti and Theileria buffeli by nucleotide sequencing and polymerase chain reaction. Molecular and bio-286 chemical parasitology, 56(1), 169-175.
Cufos, N., Jabbar, A., de Carvalho, L. M., & Gasser, R. B. (2012). Mutation scanning-based analysis of Theileria orientalis populations 261 in cattle following an outbreak. Electrophoresis, 33(13), 2036-2040.
LL: Thank you. Papers by Taylor and Kawazu were cited in the Table 1. Paper by Cufos has now been added in the text for the mixed infections discussion.
- Lines 41-44: Certain relevant references are missing in the introduction and discussion. Particularly those papers that reported first outbreaks in other states in Australia and the first discovery of certain T. orientalis genotypes.
E.g., Gebrekidan, H., Gasser, R.B., Perera, P.K., McGrath, S., McGrath, S., Jabbar, A., 2015. Investigating the first outbreak of oriental theileriosis in cattle in South Australia using multiplexed tandem PCR (MT-PCR). Ticks and Tick-borne Diseases 6, 574-578.
LL: Thank you for the feedback. References have been added.
- Since the sampling was restricted to the south-west region of WA, authors can consider including that to the title rather than 'Western Australia'.
LL: Thank you for this feedback. This has been modified.
- Line 85: Why didn't the authors use a study which investigated the prevalence of Theileria orientalis in WA or any other state to calculate the sample size for the current study?
LL: The samples were collected as a part of a broader project. Therefore, there was a limitation on the sample size.
- Figure 3. Phylogeny of bovine Theileria orientalis genotypes...
LL: Thank you. This has been modified.
- Consider stating the location of each sequence in Figure 3 rather than the isolate name.
LL: Thank you for this feedback. The label of the location (F5, F12, F15, F24 and F29) of each sequence have been added.
- Line 187: Is it across WA or south-west region of WA?
LL: South-west region of WA. This has been modified.
- Line 209: Babesia bigemina. Please check spellings.
LL: Thank you. This has been corrected.
- Lines 218-220: Add references.
LL: Thank you for this feedback. A reference has been added.
- Can authors discuss why mixed infections were not detected? Mixed infections of 2 or more genotypes of T. orientalis have been commonly reported in other states of Australia.
LL: Thank you for this feedback. This has been added in the discussion. Mixed infections of two or more genotypes of T. orientalis has commonly been detected in NSW, Qld, Vic and SA. Considering the detections of two genotypes of T. orientalis in the present study, there might be a possibility of potential mixed infections in WA. However, mixed genotype populations were not able to be detected using cPCR and Sanger sequencing in the present study.
Reviewer 2 Report
Thank you for the opportunity to review “Distribution and prevalence of Theileria orientalis genotypes in adult lactating dairy cows in Western Australia” by Leong et al. This manuscript describes a surveillance study of Theileria orientalis in regions of south-western Australia. The authors describe PCR detections of genotype Ikeda at low prevalence and the first detection of genotype Buffeli in the state of Western Australia, where previously only Ikeda had been detected.
The study is small, but representative of T. orientalis distribution in WA. I don’t see any major issues regarding the methods and statistical analyses. The authors should be congratulated for including confidence intervals in their results.
Line comments:
Table 1: Four PCR primer pairs are listed here but only results from two (MPSP long and short) PCRs are reported in results.
Lns 153-154: Is there a good reason for the increased sensitivity observed? Only one sample difference from my read. Was that sample low titre and/or DNA heavily degraded (DNA fragmentation may cause longer PCR reactions to fail)?
Table 2: It’s well reported that Theileria orientalis infections are commonly observed as mixed populations of different genotypes. This may be less of a problem in Western Australian samples as only Ikeda had been observed previously but this study did observe the Buffeli genotype present. Sanger sequencing would generally not show mixed genotype populations but there may be hints shown by mixed base calls in Sanger chromatograms or failed sequencing results for particular samples. Were there any indications that some of the samples had more than one genotype present?
Lns 164-165 and Figure 2: I’m not sure that this figure fits into this manuscript. It is an interesting result, and this may be true for “pure” Ikeda and Buffeli infections but any mixture of these genotypes, or the addition of those not previously detected in WA, would likely be very difficult to separate via melt-curve analysis. If this is retained, a caveat should at least be noted (maybe in the discussion) that mixed infections would be difficult or impossible to call based on this method.
Figure 3: Could the authors please check their GenBank accessions. The bolded labels in Figure 3 refer to accessions KM624621.1 and AP011946.1. My searches of NCBI Nucleotide for these accessions produce the Buffeli sequence from Bogema et al 2015 and the chromosome 1 of the Shintoku genome sequence, respectively.
Ln 195: Is it possible to include locations for these on the map in Figure 1? I’m not sure that international readers would be familiar with local government regions of WA.
Ln 219: Is distribution of the vector another factor? Does H. longicornis prefer certain regions in WA? Previous distribution maps I have seen appear to show a preference for the south coast of WA. Is weather a factor? Tick prevalence may be related to periods of high or low rainfall?
Ln 224-229: There are several scenarios that are linked to asymptomatic Ikeda infection. See Hammer et al 2016 Parasit & Vectors 9:34 http://doi.org/10.1186/s13071-016-1323-x and Emery 2021 Pathogens 10:1153 https://doi.org/10.3390/pathogens10091153 for some examples.
Discussion general: It would be good to have some discussion of potential for mixed infections, especially considering the Buffeli genotype detection.
Author Response
Reviewers 2
Comments and Suggestions for Authors
Thank you for the opportunity to review “Distribution and prevalence of Theileria orientalis genotypes in adult lactating dairy cows in Western Australia” by Leong et al. This manuscript describes a surveillance study of Theileria orientalis in regions of south-western Australia. The authors describe PCR detections of genotype Ikeda at low prevalence and the first detection of genotype Buffeli in the state of Western Australia, where previously only Ikeda had been detected.
The study is small, but representative of T. orientalis distribution in WA. I don’t see any major issues regarding the methods and statistical analyses. The authors should be congratulated for including confidence intervals in their results.
Line comments:
Table 1: Four PCR primer pairs are listed here but only results from two (MPSP long and short) PCRs are reported in results.
LL: Thank you for the feedback. The reason for not reporting the 16S and 18S results is because 16S was used for assessing the quality of the DNA extractions and 18S was used for narrowing down the Piroplasmida organisms to ensure the accuracy of the MPSP PCRs. This has been added in the methods.
Lns 153-154: Is there a good reason for the increased sensitivity observed? Only one sample difference from my read. Was that sample low titre and/or DNA heavily degraded (DNA fragmentation may cause longer PCR reactions to fail)?
LL: Thank you for this feedback. This has been removed.
Table 2: It’s well reported that Theileria orientalis infections are commonly observed as mixed populations of different genotypes. This may be less of a problem in Western Australian samples as only Ikeda had been observed previously but this study did observe the Buffeli genotype present. Sanger sequencing would generally not show mixed genotype populations but there may be hints shown by mixed base calls in Sanger chromatograms or failed sequencing results for particular samples. Were there any indications that some of the samples had more than one genotype present?
LL: Thank you for this feedback. This has been added in the discussion.
Lns 164-165 and Figure 2: I’m not sure that this figure fits into this manuscript. It is an interesting result, and this may be true for “pure” Ikeda and Buffeli infections but any mixture of these genotypes, or the addition of those not previously detected in WA, would likely be very difficult to separate via melt-curve analysis. If this is retained, a caveat should at least be noted (maybe in the discussion) that mixed infections would be difficult or impossible to call based on this method.
LL: Thank you for this feedback. This has been added in the discussion.
Figure 3: Could the authors please check their GenBank accessions. The bolded labels in Figure 3 refer to accessions KM624621.1 and AP011946.1. My searches of NCBI Nucleotide for these accessions produce the Buffeli sequence from Bogema et al 2015 and the chromosome 1 of the Shintoku genome sequence, respectively.
LL: Thank you. My searches are the same as yours. Shintoku was a former name of Ikeda in Japan.
Ln 195: Is it possible to include locations for these on the map in Figure 1? I’m not sure that international readers would be familiar with local government regions of WA.
LL: Thank you. The locations were included in the figure title.
Ln 219: Is distribution of the vector another factor? Does H. longicornis prefer certain regions in WA? Previous distribution maps I have seen appear to show a preference for the south coast of WA. Is weather a factor? Tick prevalence may be related to periods of high or low rainfall?
LL: Thank you for this feedback. This has been added in the discussion. The distribution of T. orientalis has been found mostly located in the coastal regions with high rainfall. The first Theileria infection in SA was recorded in a high rainfall area of SA in 2012 with minor BATOG outbreaks occurring in 2014, 2016 and 2017. An increasing H. longicornis population size has been reported due to elevated rainfall especially in the southeast region of SA, resulting in widespread BATOG in SA. H. longicornis, which were identified in the BATOG outbreaks in Albany, indicating H. longicornis are the common vector in Albany and Demark shires, which are geographically close to each other. H. longicornis ticks were also reported in Harvey shire. However, the geographical distribution of H. longicornis ticks in WA is lacking.
Ln 224-229: There are several scenarios that are linked to asymptomatic Ikeda infection. See Hammer et al 2016 Parasit & Vectors 9:34 http://doi.org/10.1186/s13071-016-1323-x and Emery 2021 Pathogens 10:1153 https://doi.org/10.3390/pathogens10091153 for some examples.
LL: Thank you for this feedback. This has been added.
Discussion general: It would be good to have some discussion of potential for mixed infections, especially considering the Buffeli genotype detection.
LL: Thank you for this feedback. This has been added. Mixed infections of two or more genotypes of T. orientalis has commonly been detected in NSW, Qld, Vic and SA. Considering the detections of two genotypes of T. orientalis in the present study, there might be a possibility of potential mixed infections in WA. However, mixed genotype populations were not able to be detected using cPCR and Sanger sequencing in the present study.